



# Effects of seasonal and diel variations in thermal stratification on phytoplankton in a regulated river

Eunsong Jung[1], Gea-Jae Joo[1], Hyo Gyeom Kim[2], Dong-Kyun Kim[3], Hyun-Woo Kim[4*]

[1]Department of Integrated Biological Science, Pusan National University, Busan, 46241, Republic of Korea
[2]The Fisheries Science Institute, Chonnam National University, Yeosu, 59626, Republic of Korea.
[3]K-water Institute, Daejeon 34350, Republic of Korea
[4]Department of Environmental Education, Sunchon National University, Suncheon, 57922, Republic of Korea

*Correspondence to*: Hyun-Woo Kim (hwkim@sunchon.ac.kr)

**Abstract** Thermal stratification in lotic systems has occurred globally and more frequently in recent decades due to global warming and artificial river modification, often with negative ecological effects. However, studies on river stratification have been restricted to rivers below dams or reservoirs affected by their water release and lacked examination of diel stratification and its impact on phytoplankton, in particular. In the present study, we assessed the degree of thermal stratification, its environmental drivers, and the response of water quality and phytoplankton assemblage against stratification in the mid-lower reach of the Nakdong River, whose morphology has been highly modified, including the construction of eight weirs. We implemented vertical temperature profiling at three study sites, both seasonally and diurnally. Then, we calculated three stratification indices: relative water column stability, Schmidt stability, and maximum temperature gradient. Three indices for assessing the degree of stratification showed that most sites experienced diel stratification during summer. Principal component analysis showed that stratification significantly led to seasonal and diel variations in the water environment. Solar radiation and air temperature were positive controllers, while a negative controller (in this case, the river flow rate) existed only for diel variation in the stratification. Higher abundance and surface cell accumulation of cyanobacteria *Microcystis* were observed at the stratified sites, and the diel variations in its biomass (chlorophyll *a*) in the surface water were primarily associated with the stratification indices instead of with other temperature and nutrient variables. Overall, the results suggest that the river has summer stratification, which is involved in amplifying cyanobacterial bloom intensity. Without a suppressing factor, summer stratification is expected to be recurrent in the river, and thus mitigating the developed stratification is needed by promptly regulating the river flow.

## 1 Introduction

Due to the increase in global temperature, thermal stratification has increased in both frequency and regional ranges in freshwater ecosystems (Pilla et al., 2021) and has become an essential determinant of the physicochemical characteristics of water bodies. This is a phenomenon wherein the waterbody stability is greater than the effect of vertical mixing due to the vertical density and temperature difference, and the entire water column does not mix. Under stratification, the distribution of materials and biochemical processes becomes uneven in depth. Increased interactions of water layers with the atmosphere and riverbed further increase depth-dependent unevenness. Stratification generally degrades aquatic ecosystems, causing harmful cyanobacterial blooms on surfaces, oxygen depletion, and nutrient elution in bottom water are the most severe stratification outcomes (Wagner and Adrian, 2011, Liu et al., 2019). Evaluating thermal stratification is urgent in possible outbreak areas to manage freshwater ecosystems.

Previously, rivers were not seriously considered in stratification research because the water flow generated sufficient turbulence. However, river alterations such as dredging and flow regulation by artificial structures, including dams and weirs, gradually increase stratification observation cases across decades (Nilsson et al., 2005). Thermal stratification of rivers has been reported in several weir pools and rivers below dams and reservoirs (Engel and Fischer, 2017; Reinfelds and Williams,





2012; Jin et al., 2019). River stratifications possess different characteristics (e.g., thermal structure, intensity, and period) than the typical lake-based stratification, which forms multiple thermal layers and temperature gradients (thermoclines) in between. Therefore, when assessing river stratification, we should apply objective stratification criteria and find an ecological response against stratification.

In South Korea, Four Major River project completed in 2011 drastically altered the rivers (Lee et al., 2021). The Nakdong

River, the second largest river in the nation, is famous for its natural sluggish riverbed and minimal slope (>1/10,000). In the project, 334 km of the total river length (525 km) has been altered by eight serial weirs and 4.4 billion m$^3$ of dredging. The average river depth increased up to 6–12 m. After the project, severe water quality deterioration and excessive cyanobacterial proliferation previously restricted to downstream areas of the river are now frequently reported in midstream areas (Park et al., 2021). These effects are often considered a consequence of the formation of stratification. However, few studies have dealt

with stratification or vertical profiles in the river, except for Kim et al. (2019).

Therefore, the purpose of this study was (i) to assess the degree of thermal stratification in the Nakdong River by applying various stratification indices, (ii) to elucidate the relationships between the stratification degree and water quality parameters, and (iii) to identify the effects of thermal stratification on phytoplankton communities.

## 2 Methods

### 2.1 Study area

The Nakdong River is located in the south-eastern part of the Korean peninsula, with a length of 511 km and a catchment area of 23,690 km$^2$ (Fig. 1). The river has a typical temperate monsoon climate and a mean annual precipitation of approximately 1500 mm. The FMR project in 2011 led to the river having a more stagnant flow regime than before. Mid-downstream of the

river was investigated, in which three sampling sites located 20, 69, and 94 km upstream from the estuarine barrage. The salinity of the downstream sampling site was below <0.2 psμ, indicating that there was no saltwater intrusion to the sampling site.

### 2.2 Sampling, sample analysis, and data collection

From 2017 to 2018, we conducted five seasonal surveys in the center of the river using a boat at the up-, mid-, and downstream sampling sites. At noon, we measured the water temperature at one-meter intervals using a portable water quality monitoring device (M-4000, Technology & Environment Corp., Seoul, South Korea). Samples for chemical and phytoplankton analyses were taken at three vertical water layers (depths of 0.5 m and 3 m from the surface water and 0.5 m above the river bottom) by using a Van Dorn sampler and samples for phytoplankton analyses were fixed with Lugol's solution. In August, diel surveys

were additionally conducted at two stratified sites for 24 hours at three-hour intervals.

Filtered water for the concentrations of nitrate (NO$_3$-N), orthophosphate (PO$_4$-P), and silica (SiO$_2$) was analyzed via spectrophotometry with glass microfiber filters (Whatman GF/C, 0.45 μm). Chlorophyll *a* concentration was determined after the quantitative concentration, following a previously published extraction method (Wetzel and Likens, 2000). The concentrations of nutrient and chlorophyll *a* were determined using a UV-VIS spectrophotometer (UV-1601, Shimadzu Corp.,

Tokyo, Japan), in accordance with standard methods for examining water and wastewater (APHA, 1995).

Phytoplankton samples from three vertical water layers were identified by light microscopic observations to the level of species and sometimes to the level of genus, based on Foged (1978), Cassie (1989), and Round et al. (1990). We examined the cell density of dominant and sub-dominant species for each of three major phytoplankton phyla: chlorophytes, bacillariophytes, and cyanobacteria. Quantitative assessment of the phytoplankton was performed by counting independent cells under ×200



and ×400 magnification using a Sedgewick-Rafter (S-R) chamber with a light microscope (Axioskop 40, Carl Zeiss Inc., Göttingen, Germany).

Other environmental data that corresponded to our survey date and time were collected. Meteorological (air temperature, irradiance, and wind speed) and hydrological (flow rate and water elevation) data were collected from the local stations of the Korean Meteorological Administration and the Nakdong River Flood Control Office (Fig. 1).


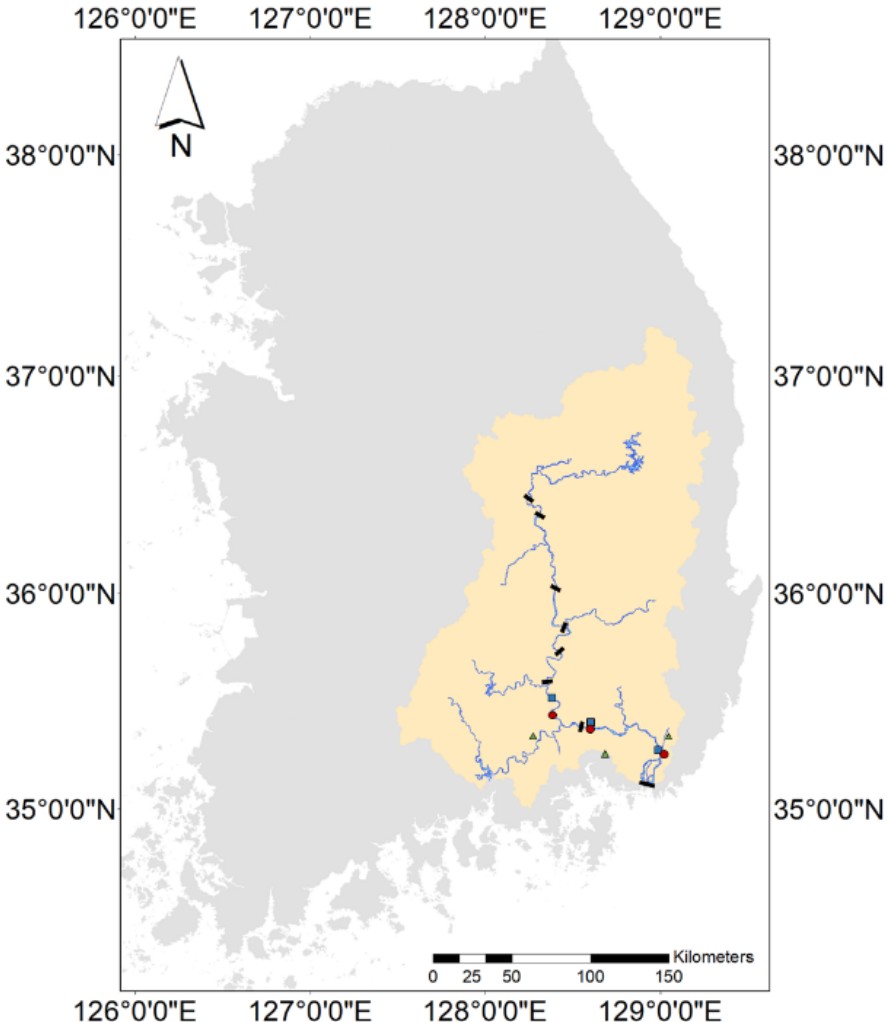

**Figure 1:** ● **Sampling site for water quality and phytoplankton, ■ hydrological monitoring station, ▲ meteorological monitoring station, ▌ weir**

**2.2 Thermal stratification indices**

We examined the existence and intensity of thermal stratification using three indices (i) RWCS, relative water column stability (Padisák et al., 2003), (ii) S, Schmidt stability (Idso, 1973), and (iii) Max, maximum temperature gradient. First, the relative water column stability is the ratio of the difference in density between the surface ($\rho_s$) and the bottom ($\rho_b$) layers to the density difference between 4 °C ($\rho_4$) and 5 °C ($\rho_5$) water, which provides an overall assessment of the mixing status along the water





column (Eq. 1). We calculated the water densities (kg m³) using a multidimensional equation of water temperature, following

Lawson et al. (2007).

$$RWCS = \frac{\rho_b - \rho_s}{\rho_4 - \rho_5},$$  (1)

Second, the Schmidt stability index (S) indicates the mixing energy demand per unit area for complete mixing (Eq. 2). We

used the rLakeAnalyzer package running on the statistical shell R (version 4.0.3) and RStudio (version 1.3.959) to calculate

the index. This package needs vectors for water temperature, cross-sectional areas, and depths of measurements. Because of

the lack of river areas across the entire depth, we assumed water columns of 1 m² area.

$$S = \frac{g}{A_s} \int_0^{z_b} (z - z_v)\, \rho_z\, A_z\, \partial_z ,$$  (2)

where $g$ is the acceleration due to gravity, $A_s$ is the surface area of the waterbody, $A_z$ is the cross-sectional area at depth $z$, $z_b$ is

the maximum depth, and $z_v$ is the depth to the center of volume. Finally, we identified the maximum temperature difference

between adjacent depths in the water column to assess the extent and location of water layer separation, which was regarded

as the thermocline in this study. To assess the formation of thermal stratification, the values of each indicator were compared

with the previously suggested criteria (Yang et al., 2016a; Padisák et al., 2003; Engelhardt and Kirillin, 2014; Yankova et al.,

2016; Sherman et al., 1998; Lampert and Sommer, 1997). Descriptive criteria are presented in Table 1.

| Index | Abbrev. | Equation | Low threshold | High threshold |
|---|---|---|---|---|
| **Relative Water Column Stability** (Padisák et al., 2003) | RWCS | $RWCS = \frac{\rho_b - \rho_s}{\rho_4 - \rho_5}$ | 30 | 50 |
| **Schmidt stability** (Idso, 1973) | S | $S = \frac{g}{A_s} \int_0^{z_b} (z - z_v)\, \rho_z\, A_z\, \partial_z$ | 30 J m⁻² | 100 J m⁻² |
| **Maximum temperature gradient** (many) | Max | Maximum slope within the water column | 0.25 °C m⁻¹ | 1 °C m⁻¹ |

**Table 1: Three stratification indices with abbreviations, equations, and low and high threshold values from literatures.**

### 2.3 Data analysis

We identified the spatio-temporal variations of the calculated stratification indices at seasonal and diel scales and presented

them in figures. Environmental and phytoplankton data sets at the two time scales were prepare to identify the environmental

drivers of stratification and the effects of stratification on the vertical distributions of water quality and phytoplankton

assemblage in the river. The environmental data sets consisted of the stratification indices, hydrological and meteorological

factors, and water quality parameters labelled with three sampling depths (0; 0.5 m, 3; 3 m, and B; bottom). Each environmental

variable except for the water quality parameters was scaled from 0 to 1 to eliminate the difference in their units of measure.

Three variables of the same water quality parameters with different depth-labels were scaled together unless their vertical

difference would be removed. The phytoplankton data sets consisted of nine variables from the three major phyla with three

depth-labels, which are the summed cell densities of their dominant and sub-dominant species. We log (x+1) transformed the





phytoplankton cell densities. Two raw data-based multivariate analyses were used to assess the relationships (i) among environmental variables and (ii) between environmental variables and phytoplankton variables. Considering the relationships would vary depending on the time scale of analyses, the seasonal and diel data sets were analysed separately.

First, principal component analyses (PCA) were used to describe correlation structure between different variables for the respective environmental data sets. For the respective PCA, we identified two environmental gradients (principal components) along which the variation in the data is maximal. PCA yielded coordinates for the respective variables against the first two PCs with Eigenvalues greater than 1. Eigenvalue > 1 indicates that PCs account for more variance than accounted by one of the original variables and is considered significant (Kim and Mueller, 1987). The percentages of explained variance by the two

PCs were presented. PCA biplots were produced using the produced coordinates and the angles among variables vectors reflect their correlation. Bartlett's test of sphericity was performed to examine the suitability of the data for PCA. All the environmental data sets showed Bartlett's significance level lower than 0.001, indicating that there were significant relationships among variables and PCA were useful.

Second, canonical correspondence analysis (CCA) was used to determine the main environment variables responsible for the

major variability trends of the river's phytoplankton assemblage. Environment variables in the environmental data sets with the same time scale were selected on the basis of the PCA results following Lopes et al., 2005. Variables considered were only those showing a significant correlation ($P < 0.05$) to PC1 and PC2. All combined variables in the PCA were eliminated from CCA since they would promote information redundancy (Hall & Smoll, 1992) and analysis distortion (Pielou, 1984). According to Ter-Braak & Prentice (1988), CCA is much more robust when just a few environment variables are needed to

identify the species distribution. Consequently, 5 and 6 environmental variables were used in present CCA at seasonal and diel scales. Before the CCA, detrended correspondence analyses (DCA) were performed to determine whether the variables in the phytoplankton data sets followed a unimodal or linear response model. All DCA had resulted first ordination axes (scaled in units of standard deviation, SD) with the length higher than 4 SD, which clearly indicated a unimodal response.

Both of PCA and CCA were followed by Monte Carlo permutation tests to test the significance level of the first two axes and

to determine the parameters that significantly contributed to the axes (Leps and Smilauer,2003).

Kruskal-Wallis ANOVA and Spearman's rank-order correlation were used to study the vertical difference and correlation structure between variables to account for non-normal distribution of water quality parameters. All statistical analyses and visualizations were performed using the FactoMineR, vegan, stats, and ggplot2 packages within R v4.0.3 (R Core Team, 2021) and RStudio (Version 1.3.959, RStudio Inc., USA).


## 3 Results

### 3.1 Stratification indices

We found that the three stratification indices had a similar seasonal variation, with the highest values in August (when it was hot), a gradual increase in May, and a drastic decrease in September (Fig. 2). Spatial variations were discernible between the

S and the other two indices. Overestimation of the stratification in the midstream by S compared to the other indices indicates that the waterbody of midstream required more turbulent energy to mix than expected for its entire stability and thermocline strength, especially in August. A strong correlation between the seasonal variations in RWCS and Max was found in our data (Spearman's $r^2 = 0.872$, $p < 0.01$). When we found stronger stratifications exceeding the thresholds of each index at the upstream and midstream, as compared to the downstream, in August, we further investigated diel variations in stratification at

the two sites.



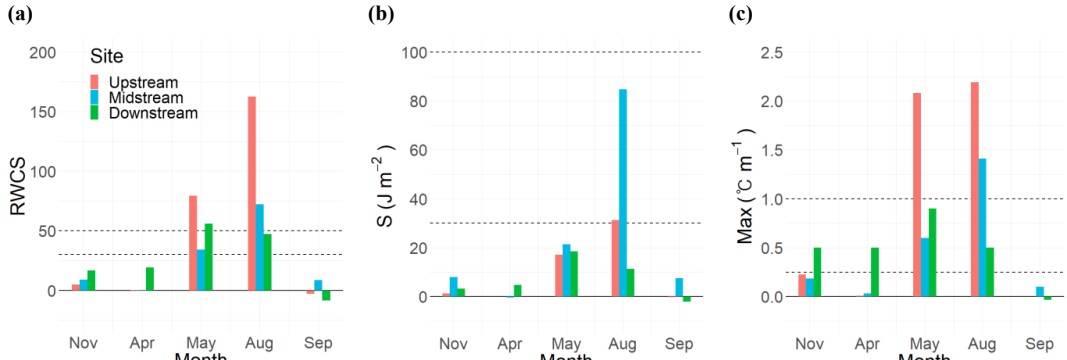

**Figure 2: Seasonal variation in three stratification indices of three sites during the study period (a: RWCS, relative water column stability, b: S, Schmidt's stability, c: Max, maximum temperature gradient, dashed lines: high and low thresholds for each index)**


The maximum S was 97.6 J m$^{-2}$ at the midstream at 15:00 in August, which was very close to the higher stratification threshold of 100 J m$^{-2}$ (Fig. 3). The midstream remained above the lower criterion (S = 30 J m$^{-2}$), but the upstream was below the criterion, except for at 12:00. For higher criteria of RWCS and Max (RWCS = 50, Max = 1 °C m$^{-1}$), the upstream was stratified for most of the day, while the midstream destratified during the night. Stratification of the upstream and midstream regions in August

was also supported by the presence of steep thermoclines (Fig. 4). In the midstream, as the surface of the waterbody warmed up, a steep thermocline developed at a depth of 3–4 m from 12:00 to 15:00. As the surface cooled and the heat transferred downwards, the thermocline became slighter and migrated deeper. A similar diel migration of the thermocline was identified upstream.

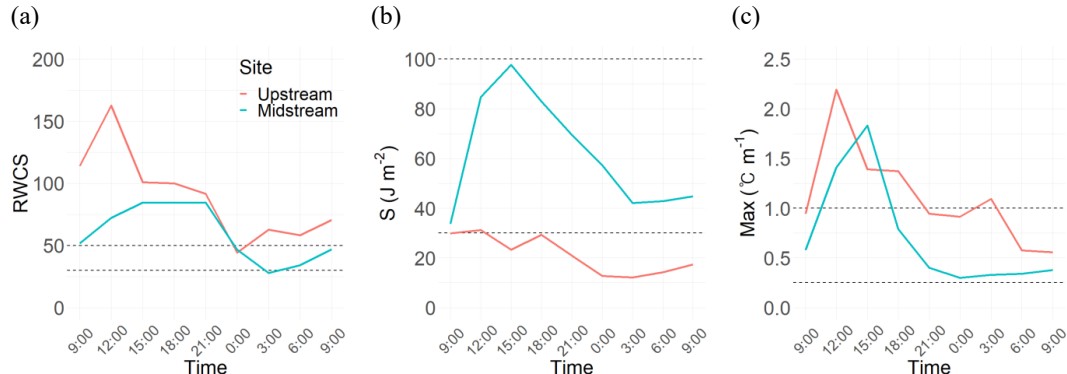

**Figure 3: Diel variations in three stratification indices of two stratified sites in August (a: RWCS, b: S, c: Max, dashed lines: high and low thresholds for each index)**



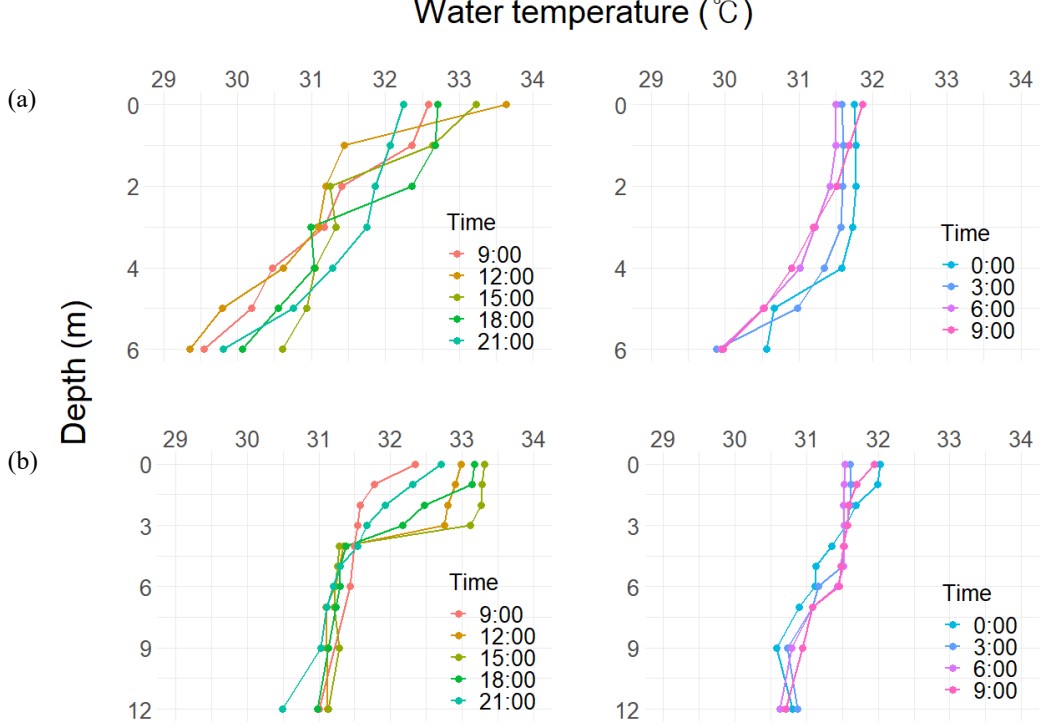

**Figure 4: Diel migration of thermoclines at two stratified sites, (a) upstream and (b) midstream in August.**

### 3.2 Environmental conditions and relationship with stratification indices

Table 2 summarizes the environmental conditions of the lower Nakdong River during the five seasons. The water temperature varied seasonally from 12.7 to 32.3°C. The river depth ranged from 6 to 12 m and was deepest in the midstream. The water level was the highest upstream, and the discharge was the highest downstream in March. The river had experienced eutrophication; the average orthophosphate and nitrate were 28.0 μg L$^{-1}$ and 2.7 μg L$^{-1}$, respectively. The mean chlorophyll $a$ value of all measurements was 34.2 mg L$^{-1}$, and summer had the highest value of 281.7 mg L$^{-1}$. Water quality variables were not significantly different among the three water depths of 0 m, 3 m, and bottom (Kruskal Wallis test, $p > 0.05$), and the seasonal PCA did not separate the water quality variables by depth in the ordination space (Fig. 5a). However, the diel survey in August showed that water temperature and chlorophyll $a$ varied significantly with depth (Kruskal Wallis test, $p < 0.01$). The diel PCA separated these variables by depth (Fig. 5b). For example, Chl was decomposed into two orthogonal vectors of Chl$_{0,3}$ and Ch$_{B}$ .




| Site | Upstream | | | | | Midstream | | | | | Downstream | | | | |
|---|---|---|---|---|---|---|---|---|---|---|---|---|---|---|---|
| Season | Nov | Apr | May | Aug | Sep | Nov | Apr | May | Aug | Sep | Nov | Apr | May | Aug | Sep |
| Maximum depth (m) | 6 | 6 | 7 | 6 | 6 | 10 | 10 | 10 | 12 | 12 | 6 | 6 | 6 | 6 | 6 |
| Water temperature (°C) | 12.7 ± 0.1 | 12.5 ± 0 | 22.3 ± 0.9 | 31.4 ± 1.2 | 23 ± 0 | 16.6 ± 0.1 | 15 ± 0 | 21.8 ± 0.4 | 32.3 ± 0.6 | 23.1 ± 0.1 | 13.9 ± 0.3 | 11.9 ± 0.4 | 20.3 ± 0.6 | 29.8 ± 0.4 | 22.6 ± 0.1 |
| $NO_3$-N (mg L$^{-1}$) | 1.6 ± 0.4 | 3.8 ± 0.1 | 3 ± 0.2 | 2.6 ± 0.1 | 3 ± 0 | 3.2 ± 0 | 2.7 ± 0.2 | 2.5 ± 0.2 | 2.1 ± 0.2 | 2.7 ± 0.1 | 2.7 ± 0 | 3.8 ± 0 | 3 ± 0.2 | 2.2 ± 0 | 2.7 ± 0 |
| $PO_4$-P (μg L$^{-1}$) | 22.2 ± 3.9 | 18.5 ± 2.1 | 7.6 ± 3.1 | 39.1 ± 9.1 | 53.9 ± 9.1 | 19.5 ± 1.7 | 8.4 ± 2.6 | 12.1 ± 1.3 | 58.6 ± 12.7 | 28.8 ± 7.3 | 30.1 ± 0.7 | 8.9 ± 3.5 | 10.7 ± 4.7 | 25.2 ± 5 | 30.1 ± 0.7 |
| $SiO_2$ (mg L$^{-1}$) | 0.4 ± 0.1 | 1.1 ± 0.1 | 0.4 ± 0 | 4.2 ± 0.2 | 6.1 ± 0.1 | 3.8 ± 0.4 | 1.6 ± 0.2 | 0.3 ± 0.1 | 3.4 ± 0.2 | 5.8 ± 0.1 | 6.4 ± 0.1 | 1.1 ± 0.7 | 0.7 ± 0.1 | 3.5 ± 0.1 | 6.4 ± 0.1 |
| Chlorophyll a (mg L$^{-1}$) | 4.6 ± 0.5 | 20.7 ± 4.8 | 6 ± 0.9 | 281.7 ± 174.4 | 10.9 ± 0.8 | 16 ± 0.2 | 26.5 ± 1.5 | 6.5 ± 0.7 | 121.5 ± 110.1 | 19.6 ± 2.9 | 10.6 ± 0.7 | 34 ± 2.4 | 4.3 ± 0.2 | 11 ± 2.6 | 8.4 ± 0 |
| Air temperature (°C) | 9.4 | 21.6 | 26.3 | 35.3 | 23.3 | 18.6 | 21 | 23.4 | 35.9 | 24.1 | 13.3 | 14.3 | 28.2 | 33 | 21.1 |
| Wind velocity (m s$^{-1}$) | 1 | 1.4 | 1.9 | 1.7 | 0.9 | 2 | 1.8 | 2.7 | 2 | 3.7 | 4.6 | 1.8 | 3.2 | 2.5 | 3.1 |
| Solar radiation (MJ m$^{-2}$) | 2.15 | 2.7 | 3.09 | 3.12 | 2.18 | 1.75 | 1.94 | 1.58 | 1.96 | 1.28 | 1.12 | 0.51 | 1.7 | 2.1 | 0.04 |
| Flow rate (m$^3$ s$^{-1}$) | -14.87 | 32.13 | 412.89 | 33.76 | 386.79 | 48.77 | 728.7 | 286.84 | 61.71 | 247.75 | 520.85 | 1,059.69 | 1,041.46 | 755.42 | 802.79 |
| Water level (m) | 3.73 | 4.82 | 4.98 | 4.87 | 4.96 | 0.65 | 0.95 | 0.75 | 0.73 | 0.99 | 0.42 | 0.64 | 0.63 | 0.5 | 0.52 |

**Table 2: Depth-averaged water characteristics and environmental factors of three sites during the study period (n = 3, mean ± S.E.).**

We assessed the relationships between various environmental variables, including the three stratification indices at two temporal scales: i) seasonal and ii) diel. The proportions of explained variance by the first two PCs for the seasonal PCA and diel PCA were 62.7 % and 67.1 %, indicating that most of the variation in the data was explained by the two PCs (Fig. 5). All stratification indices had significant contributions to both ordinations ($P < 0.05$), indicating stratification as a major driver of the water environment (Table 3). Some environmental variables, such as FR, WL, WV, and $Chl_{3,B}$ significantly contributed

only to the diel PCA.
      In the seasonal PCA, observations of the water environment were separated more by season than by site (Fig. 5a). Water environments of the three sampling sites were clearly separated mainly along the PC2, while both the PC1 and PC2 were involved in the separation of the five seasons. The PC2 was associated with RWCS, Max, SR, and $SI_{0,3,B}$ and the PC1 was associated with S, AT, $WT_{0,3,B}$, $Chl_0$, $PO_{0,3,B}$ at $P < 0.05$. Furthermore, there were strong relationships among the environmental

variables. RWCS and Max had a positive correlation with SR. S was positively correlated to $Ch_0$ and AT. Some water quality parameters such as SI, PO, and WT had a correlation among the values measured at different depths. Finally, S, Max, $SI_3$, $PO_3$, and $WT_0$ were selected for CCA considering the interrelated variables in the PCA.
      In the diel NMDS, upstream and midstream were clearly separated along the PC1, associated with S, WL, $WT_{3,B}$, $PO_B$, $SI_{0,3,B}$, and $Chl_B$, and these variables were highly site-dependent (Fig. 5b). Each site has a diel cycle driven by the PC2, associated

positively with Max, RWCS, SR, WV, AT, $WT_0$, and $Chl_{0,3}$, and negatively with FR. In particular, $Chl_{0,3}$ had a positive correlation with RWCS and a negative correlation with FR. S, AT, Max, WL, FR, and $SI_3$ were selected for CCA.

(a)                                                            (b)





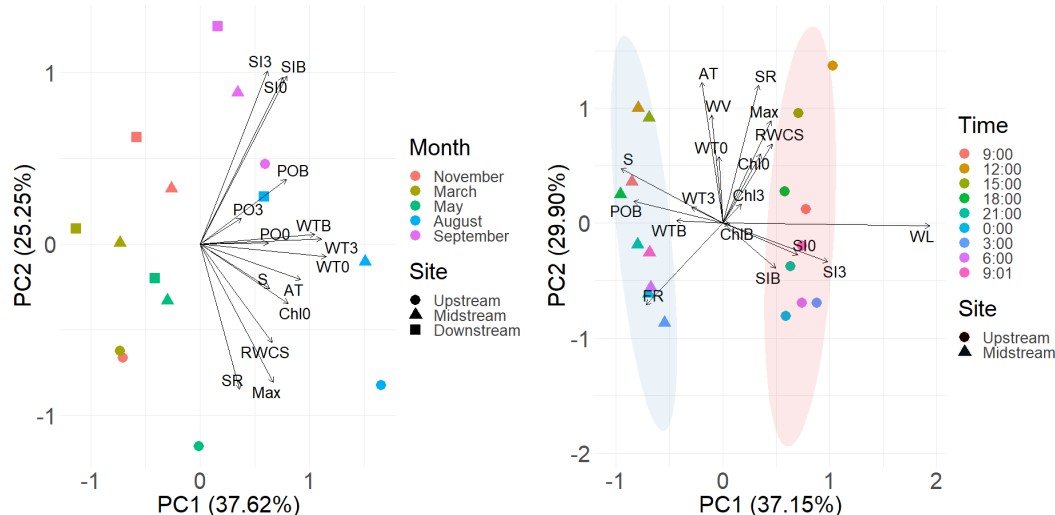

**Figure 5: Principal component analysis ordinations of the relationships among the environmental variables at (a) seasonal and (b)**
**diel scales (n = 15: seasonal, 18: diel). The environmental variables, marked with measured water depth (0, 3, and B), are as follows:**
**AT, air temperature; Chl, chlorophyll a concentration; FR, flow rate; Max, maximum temperature gradient; NO, nitrate; PO,**
**orthophosphate; S, Schmidt's stability; SI, silicate; SR, solar radiation; WL, water level; WT, water temperature; WV, wind velocity.**
**Variables displayed at *P* < 0.05.**

| | Seasonal | | | | Diel | | | |
|---|---|---|---|---|---|---|---|---|
| | PC1 | PC2 | r² | *P*-value | PC1 | PC2 | r² | *P*-value |
| RWCS | 0.684 | **-0.729** | 0.729 | 0.002 | 0.515 | **0.857** | 0.758 | 0.001 |
| S | **0.895** | -0.447 | 0.501 | 0.011 | **-0.874** | 0.486 | 0.777 | 0.001 |
| Max | 0.561 | **-0.828** | 0.682 | 0.003 | 0.414 | **0.910** | 0.779 | 0.001 |
| AT | **0.964** | -0.268 | 0.699 | 0.002 | -0.144 | **0.990** | 0.921 | 0.001 |
| SR | 0.330 | **-0.944** | 0.743 | 0.002 | 0.242 | **0.970** | 0.864 | 0.001 |
| WV | -0.102 | 0.995 | 0.285 | 0.141 | -0.103 | **0.995** | 0.694 | 0.001 |
| FR | -0.641 | 0.768 | 0.252 | 0.184 | -0.672 | **-0.741** | 0.593 | 0.002 |
| WL | 0.246 | -0.969 | 0.413 | 0.052 | **1.000** | -0.014 | 0.970 | 0.001 |
| WT0 | **0.997** | -0.079 | 0.875 | 0.001 | -0.060 | **0.998** | 0.848 | 0.001 |
| WT3 | **1.000** | 0.031 | 0.893 | 0.001 | **-0.885** | 0.466 | 0.415 | 0.015 |
| WTB | **0.998** | 0.064 | 0.878 | 0.001 | **-0.998** | 0.057 | 0.734 | 0.001 |
| Chl0 | **0.884** | -0.467 | 0.658 | 0.003 | 0.468 | **0.884** | 0.538 | 0.002 |
| Chl3 | 0.828 | -0.561 | 0.361 | 0.066 | 0.686 | **0.728** | 0.646 | 0.001 |
| ChlB | -0.949 | -0.315 | 0.082 | 0.619 | **0.993** | -0.122 | 0.471 | 0.005 |
| NO0 | -0.977 | -0.215 | 0.143 | 0.376 | 0.914 | -0.406 | 0.179 | 0.251 |
| NO3 | -0.993 | -0.116 | 0.231 | 0.184 | 0.653 | 0.757 | 0.267 | 0.108 |
| NOB | -0.806 | 0.593 | 0.136 | 0.4 | -0.546 | 0.838 | 0.022 | 0.855 |
| PO0 | **1.000** | 0.009 | 0.673 | 0.002 | -0.547 | 0.837 | 0.254 | 0.109 |
| PO3 | **0.902** | 0.432 | 0.552 | 0.007 | -0.869 | 0.495 | 0.147 | 0.325 |
| POB | **0.864** | 0.503 | 0.629 | 0.004 | **-0.969** | 0.246 | 0.533 | 0.004 |
| SI0 | 0.536 | **0.844** | 0.862 | 0.001 | **0.917** | -0.400 | 0.881 | 0.001 |
| SI3 | 0.447 | **0.894** | 0.803 | 0.001 | **0.934** | -0.356 | 0.888 | 0.001 |
| SIB | 0.551 | **0.835** | 0.878 | 0.002 | **0.753** | -0.658 | 0.744 | 0.001 |





**Table 3: Pearson correlation coefficient of determination (r² ) and *P*-values based on random permutations between the environmental variables and the PC coordinates at seasonal and diel scales (n = 15: seasonal, 18: diel). The bold values statistical significance (*P* < 0.05).**

### 3.3 Phytoplankton assemblage and relationship with stratification indices

A total of 61 phytoplankton species were identified, of which chlorophytes, cyanobacteria, and bacillariophytes accounted for 26, 20, and 15 species, respectively (Table S1). The average number of species in a water column (chlorophyte: 3.3 ± 0.2, cyanobacteria: 3.1 ± 0.2, bacillariophyte: 3.5 ± 0.2; *n* = 15) indicates that species dominance was highly similar between the water layers in the water column. Interestingly, a small number of phytoplankton cells (6806 ± 1554.5 cells ml⁻¹; *n* = 15) was detected in the bottom waters, deeper than the general euphotic depth of the river. The phytoplankton assemblage showed a
seasonal shift at the phylum level (Fig. 6). In August, the highest cell densities were observed, and the phylum Cyanobacteria dominated the assemblage. In May, Bacillariophytes dominated, and in March-April and September, co-domination of Bacillariophytes and Chlorophytes was detected. Massive cyanobacterial blooms were observed in November, August, and September, with high cell densities concentrated in the surface waters. The blooming genera varied by season, with *Microcystis* for August, the strongest stratification period, and *Aphanizomenon* for November and September, when the waterbodies mixed
well and their proliferation was independent of stratification (Table S1). Unlike cyanobacteria, the cell densities of bacillariophytes and chlorophytes had less vertical variation.

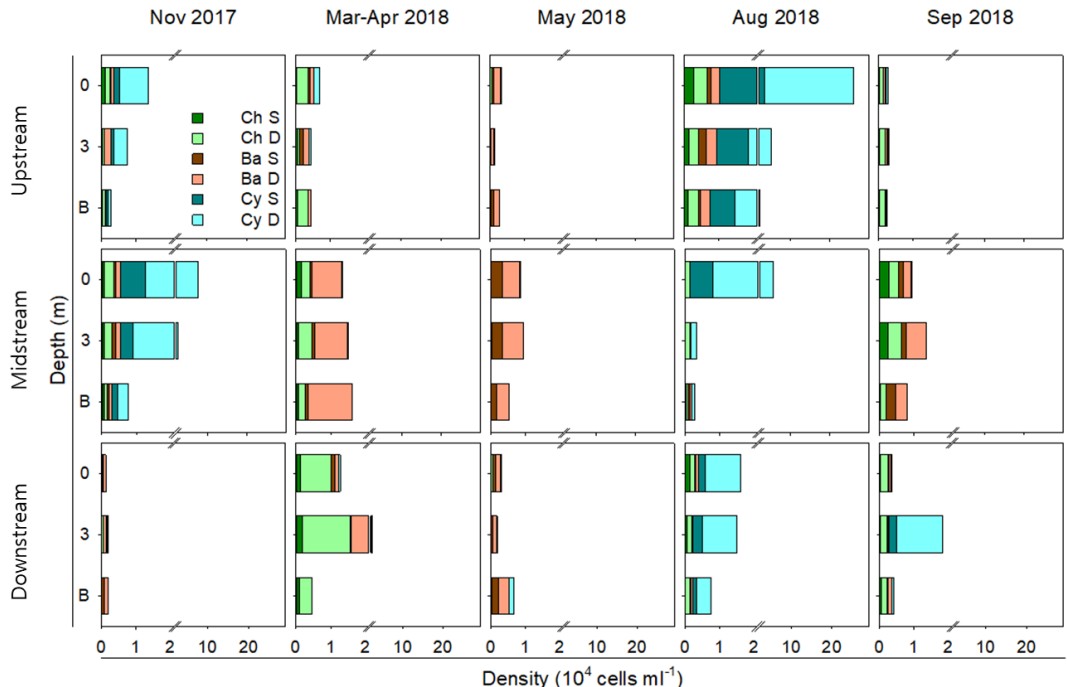

**Figure 6: Vertical distributions of cell densities for dominant and subdominant species from Chlorophyta, Bacillariophyta and**
**Cyanobacteria (i.e., Ch S: subdominant chlorophyte species)**

The significance levels of the seasonal and diel CCA models with the vertical structure of phytoplankton assemblages as response variables and the investigated environmental properties as constraining variables were 0.613 and 0.001, respectively (Fig. 7). All of the environmental variables could explain 31.0% and 72.6% of the variation in the assemblage-environment





relationship for both CCAs. Variations of phytoplankton assemblage in the diel CCA were dependent on the values of S (F =
9.65, p = 0.002), Max (F = 6.43, p = 0.005), and WL (F = 8.45, p = 0.003), but there were no significant variables in the
seasonal CCA ($WT_0$: F = 1.71, p = 0.169; $PO_3$: F = 1.31, p = 0.255). In the seasonal CCA, S and $PO_3$ had a positive relationship
with cya0 and a negative relationship with bac0,3,B. In the diel CCA, AT had a positive relationship with cya0,3,B and a
negative relationship with bac3.


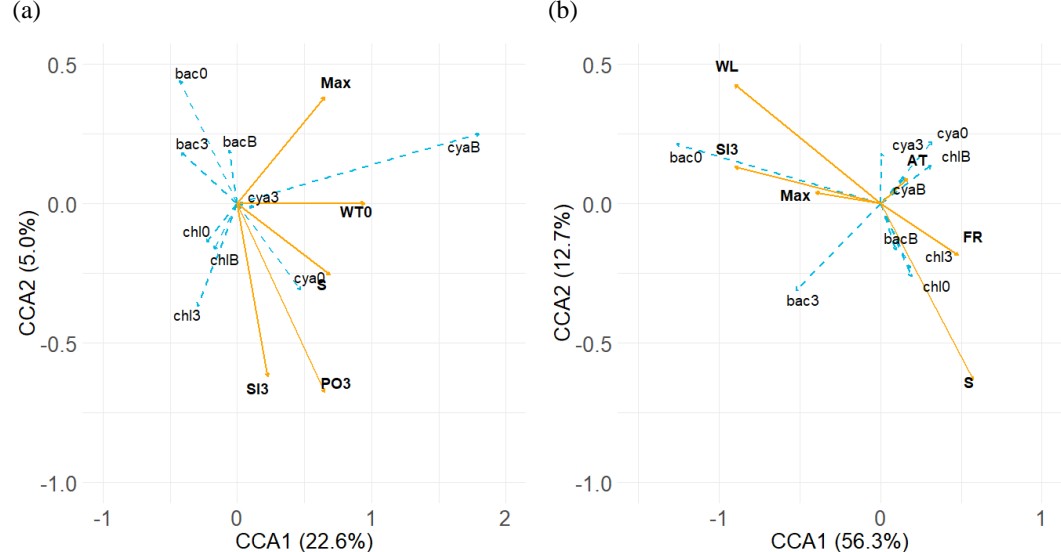

**Figure 7: Biplot diagram for the canonical correspondence analysis of the relationship between phytoplankton phyla (blue dashed line) and constraining environmental variables based on the PCA results (yellow solid line). The phytoplankton phyla and environmental variables, marked with measured water depth (0, 3, and B), are as follows: chl, Chlorophyta cell density; bac, Bacillariophyta; cya, Cyanophyta; AT, air temperature; FR, flow rate; Max, maximum temperature gradient; PO, orthophosphate; S, Schmidt's stability; SI, silicate; WL, water level; WT, water temperature.**

### 3.4 Vertical variation of Microcystis under stratification

During summer, *Microcystis wesenbergii* dominated all the study sites (Fig. 6 and Table S1). However, their surface (0.5 m
depth) cell densities and a high relative abundance at the surface layer coincided with the spatial variation of stratification
and were much higher in the stratified upstream (22.55 x $10^4$ cells $ml^{-1}$, 83.6 %) and midstream (5.44 x $10^4$ cells $ml^{-1}$, 95.2 %).
In the relatively mixed downstream, the species showed lower density (1.02 x $10^4$ cells $ml^{-1}$) and surface accumulation
(50.6 %). An identical vertical distribution pattern was observed for the subdominant cyanobacteria species *M. aeruginosa*
and *M. sp*. For comparisons with the genus, in August, five species from Bacillariophytes and Chlorophytes were
simultaneously detected at the three vertical water depths in both stratified and unstratified sites. They showed much less
vertical variation in their cell densities in both stratified and unstratified sites with low relative standard deviations (0.216 $\pm$
0.082; n = 5). In addition, they were not concentrated in the surface water (1985 $\pm$ 539 cells $ml^{-1}$, 38.2 $\pm$ 2.8 %; n = 5).

Most of the cell densities were from cyanobacteria in the summer and a strong association was found between chlorophyll *a*
at the surface and 3 m depth and the diel variations of two stratification indices, RWCS and Max, as shown in Fig. 5b. This
allowed us to further explore how diel stratification affects cyanobacterial distribution. When strong thermoclines (> 1°C $m^{-1}$)
developed in the water columns (Fig. 4), chlorophyll *a* in both stratified sites showed its maximum in the surface waters and
minimum in the bottom waters (Fig. 8). In the upstream, a shallow thermocline floated from 1-2 m depth to the water surface,





and then chlorophyll *a* became concentrated in the upper water from 9:00 to 12:00. The high surface concentrations of chlorophyll *a* rapidly decreased as the thermoclines deepened.


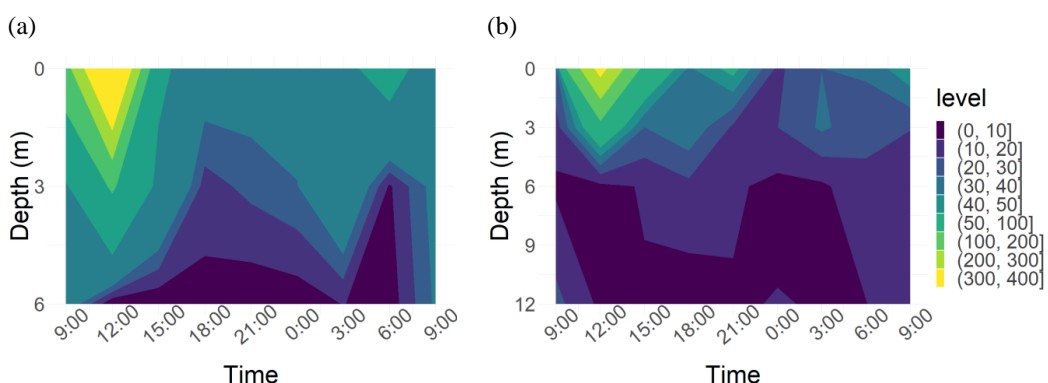

Figure 8: Diel variations of chlorophyll *a* concentration at two stratified sites, a) upstream and b) midstream in August.

## 4 Discussion

### 4.1 Characteristics of stratification in the Nakdong River

Schmidt stability (S) showed a different spatial variation from the other two stratification indices because of the dependence on the water depth of its equation. This indicates that the choice of indices can influence the stratification evaluation when multiple sites are included. Yang et al. (2016a) utilized S and relative water column stability (RWCS) for long-term monitoring of a lake, and their interchange was possible. The maximum thermocline gradient (Max) can directly represent the vertical partitioning of the water layers, which is important for the distribution and transfer of materials and thus may be the primmest

term for assessing stratification independent of spatial heterogeneity. RWCS does not require an entire temperature profile, and its simplicity allows many phytoplankton researchers to explore its role in phytoplankton ecology (Becker et al., 2008 and 2009; Cui et al., 2021 and 2022; Rocha et al., 2019). Our finding of similarity between Max and RWCS in seasonal and diel variations may provide a bridge between studies using these indices.

Thermal stratification has become a general phenology of regulated rivers and reservoirs in the global trend of river regulation

(Engel et al. 2017, Cheng et al. 2020, Xu et al. 2021). However, the stratification impact can be expressed differently depending on where it occurs and what ecological characteristics it has. The intensity of the summer stratification in the river was weak (mean: RWCS 74.3, S 41.5 J m$^{-2}$, Max 0.9 °C m$^{-1}$) compared to those of a temperate lake with a maximum depth of 21 m (mean S 92 J m$^{-2}$; Yang et al., 2016b) and many deep Arctic lakes at the beginning of the warming season (St = 100 J m$^{-2}$; Wang et al., 2020). Due to its weak intensity, river stratification seems to form a single thermocline at shallow depths smaller

than 60% of the river depth (Gorham and Boyce, 1989). Similar to other studies on unstable stratification, the thermocline exhibited diel migration and night-time destratification, as seen in other research into unstable stratification (Hare and Carter, 1984; Stauffer, 1992). This diurnal thermocline is found in warm polymictic lake systems (Bartosiewicz et al., 2019) and developed on top of the persistent thermoclines in the case of multiple thermoclines (MacKinnon and Herbert, 1996). The thermocline in the river could be differentiated from lake stratification because it moved along the entire water column and

reached the riverbed by diel migration. These features of stratification and their diel variations make them a major driver shaping water quality and phytoplankton communities in freshwater ecosystems, especially by controlling the mixing depth,



light availability, and mixing time interval, which hinders cell accumulation in the surface layer during the daytime (Jensen et al., 1994; Webster, 2000).

## 4.2 Environmental drivers of stratification

The PCA ordinations revealed that thermal stratification is one of the most important drivers of water environments in the Nakdong River, largely accounting for their seasonal and diel variations. Air temperature and hydraulic residence times are well-known major drivers of stratification (Mosley, 2015). In the Nakdong River, solar radiation and air temperature were the major drivers of stratification, increasing the surface water temperature and resulting in vertical temperature differences. The strong association of the stratification indices with air and water temperature and solar radiation coincides with previous

findings of strong relationships between these variables (Liu et al., 2019, Li et al., 2018). The dependence of the variations in stratification on these meteorological variables supports the current increasing trend of its occurrence and intensity in a changing climate (Carey et al., 2012). It is problematic that there is currently no suppressing factor against the seasonal variation of stratification in our study sites, although we tested commonly known factors inducing vertical mixing of the water bodies, such as flow and wind velocity (Stefan et al., 1996, Woolway et al., 2017). In sum, intensification of summer

stratification is inevitable in the river. The strong relationships between flow rate, the stratification indices, and chlorophyll $a$ concentration from the diel-scale analysis suggest that the possibility of flow management from relevant research (Mitrovic, Hardwick, & Dorani, 2011; Reinfelds and Williams, 2012) could be adopted to mitigate the adverse impact of stratification on the river ecosystem by operating weirs. This is supported by the discharge-growth hypothesis (Webster et al., 2000), which depends on three components: the relationship between discharge and stratification, the relationship between the vertical

distribution of phytoplankton and stratification, and the competitive advantage of a buoyant population under stratified conditions. Moreover, the hydrological factors and wind velocity exhibited greater spatial variability, indicating spatial heterogeneity in susceptibility to thermal stratification formation. This makes the upstream site more favourable for stratification in terms of its lower flow rate, lower wind velocity, and higher water level.

## 4.3 Stratification effect on phytoplankton assemblage

A phylum-level shift between seasons, the major variation in the phytoplankton assemblage, was not associated with the stratification indices but rather with orthophosphate concentration. Impacts of thermal stratification on their assemblage, which was obvious for buoyant cyanobacteria in the river, could be selective depending on phytoplankton groups which possess different adaptations and functional traits (Reynolds, 2012). Unlike cyanobacteria, the vertical cell distribution of chlorophytes and bacillariophytes remained uniform over five seasons, even during the stratification period, providing further information

on stratification. In the absence of sufficient turbulent mixing, most phytoplankton species which do not have a buoyancy regulation sink (Huisman et al., 2002). Under stratification, deep-water populations of these phyla are affected more by nutrients and light availability than by water stability itself (Becker et al., 2009; Zhang et al., 2010). Our stratification was too unstable and brief to have a vertical gradient of nutrients and insufficient for these phyla to possess any vertical distribution. In addition, blooms of the cyanobacteria *Aphanizomenon* during fall-winter tolerant for water temperature in the range of 5–

15 ℃ in the river (Ryu et al., 2016) hindered the detection of a significant relationship between stratification and cyanobacterial density in a seasonal scale. Thus, summer period and cyanobacteria are the high-priority in determining the effects of river stratification on phytoplankton.

   Two harmful cyanobacterial species, *Microcystis wesenbergii* and *M. aeruginosa*, were enhanced by summer stratification in terms of their abundance and cell accumulation in the surface water. They are two of the most common bloom-forming

*Microcystis* species in many countries, including the Nakdong River, and dominate successively from summer to autumn (Imai et al., 2009; Ozawa et al., 2005; Park et al., 1993, 1998). In particular, monocultures of large colonies of *Microcystis* are explained by buoyancy control, which accommodates diel fluctuations in stratification and mixing in low-latitude lakes



(Reynolds et al., 2002). Diel fluctuations in their biomass with high values of chlorophyll *a* in the surface water which did not persist as the thermocline weakened and deepened were primarily associated with stratification indices compared to other variables, such as water temperature and nutrients (Fig. 5b). According to Webster et al. (2000), whether or not the river water column is mixed at least once on a diurnal basis has great significance for the distribution of slowly floating Cyanobacterial genera, *Anabaena*. If the population was uniformly dispersed through the water column at sunrise, then during daylight hours, the population would not accumulate significantly into the near-surface euphotic zone where photosynthesis occurs. However, several studies have reported that *Microcysti*s colonies remain largely in the epilimnion during the summer, where they take up nutrients (Reynolds et al., 1981; Ibelings et al., 1991). Cyanobacteria, including *Microcystis*, have enhanced growth rates under stratification (O'Neil et al., 2012) and often form massive blooms as a result. Before the river was modified in 2011, Ha et al. (2000) had reported an accumulation of the same genus on the surface water during a calm night in a downstream of the river and this could explain the intensification of cyanobacterial proliferation in the river after the river modification. Thus, we suggest that stratification involves maintaining cyanobacteria in surface water once they occur and amplifying the bloom intensity.

## 5. Conclusion

Our study analyzed the degree of thermal stratification, its environmental drivers, and the response of water quality and phytoplankton assemblage to stratification in a river after intensive river channel modification, including the construction of eight weirs. Three indices for assessing the degree of stratification showed that most of the sites experienced diel stratification during summer. The PCA results showed that stratification led to significant seasonal and diel variations in the water environment. Unlike other phytoplankton phyla, buoyant cyanobacteria changed their cell density vertically in the water column. Blooms of two cyanobacterial genera were observed, and the stratification effect on *Microcystis* was further assessed as it bloomed during the stratification period. Higher abundance and surface cell accumulation of the genus were observed at the stratified sites, and the diel variations in its biomass (chlorophyll *a*) in the surface water were primarily associated with the stratification indices compared to other temperature and nutrient variables. Thus, we suggest that stratification involves maintaining cyanobacteria in surface water once they occur and amplifying the bloom intensity. With respect to the seasonal and diel variations in stratification, solar radiation and air temperature were the positive controllers, while a negative controller (river flow rate) existed only for diel variation in the stratification. Without a suppressing factor, summer stratification is expected to be recurrent in the river, and thus mitigation of the developed stratification is needed by promptly regulating river flow. Future research will extend this study by focusing on the stratification duration and destratification interval during the entire summer, as well as its ecological impact.

### Data Availability

Data are available from the authors upon reasonable request

### Author Contributions

EJ conceived the study, EJ and HK conducted field surveys and analyses, EJ generated figures and text, and HK, DK, GJ and HK* revised figures and text.



**Competing Interests**

The authors declare that they have no known competing financial interests or personal relationships that could have appeared to influence the work reported in this paper.

**Acknowledgements**

This research has been performed as Project No. Open Innovation R&D (2021-TE-RR-95-1715) supported by K-water (Korea Water Resources Corporation).

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
