# Peer review of "Effects of seasonal and diel variations in thermal stratification on phytoplankton in a regulated river"

_Biogeosciences, 2022_

## Author Comment (AC1)

**Response to referee #1 (our answers in blue):**

**General comments**

The authors present an interesting manuscript about the different effects of seasonal and diel variations in a river's thermal stratification on phytoplankton community. This work is timely given the recent intensification of interest in global temperature increase and helps predicting its short/long term consequences in freshwater ecosystems. I am pleased that the authors made a valuable contribution to the field with the high-frequency data of phytoplankton community rarely seen in other studies. The authors hypothesized that river stratification would have different environmental drivers and effects on phytoplankton in the two different time scales then analyzed them separately. Interestingly, the authors found that the seasonal shifts in phytoplankton community structure were either insensitive and showed a limited response to the stratification indices. Summer cyanobacterial bloom intensity, here cell abundance and accumulation into the surface water, was positively affected by the diel variations in the stratification indices and thermocline. Based on the environmental drivers of stratification, the authors discuss the generalization of stratification events for the river system and the implementation strategy for flow management to mitigate cyanobacterial blooms. Overall, the manuscript has interesting research questions and the data collection/analysis/interpretation seem sound. However, the manuscript needs to be revised before publication. I hope the comments below can help the authors improve their manuscript.

We would like to thank Reviewer#1 for the positive and constructive review. We have carefully responded to each comment/suggestion and did our best to improve the manuscript accordingly. We clarified and simplified the paper all along: We re-wrote the methods section to make it easier for the reader to follow. The diel stratification was discussed more thoroughly. All co-authors edited the letter and the manuscript, with additional English editing by a native speaker collaborator. We believe that the paper flows better now, making it easier for the reader.

**Specific comments**

Section 2.2: the used thermal stability indices are all based on the vertical temperature difference (potential energy), but their calculations lack the concept of vertical mixing (mixing energy) that against the formed stratification. Prandtl number, Richardson number, or Lake number could be additionally considered when appropriate (Kirillin and Shatwell, 2016).

We admit that the stratification indices we used do not include a term for vertical mixing as mentioned by the reviewer. However, we will keep the indices as they have advantage over the suggested indices after considering two major issues written below. We will add a clarification for the choice of indices.

First, one of the purposes of the study is to scale the river stratification then test its relationship with hydrological and meteorological conditions. The indices used in the present study (RWCS, Schmidt stability, and maximum temperature gradient) do not include hydrological nor meteorological parameters in their calculation unlike the

suggested indices. Also, we found that most researches on phytoplankton response against stratification in freshwater ecosystems had used these indices (Becker et al., 2008 and 2009; Cui et al., 2021 and 2022; Rocha et al., 2019). In the Discussion part, we compared our findings with other stratification cases based on these indices.

Second, thermal structure is always "subjectively" identified because an optimal solution for the threshold determination is rarely found (Zhang et al., 2014a, Chu and Fan, 2011). A number of studies have evaluated the overall thermal structure to study its response to external forces, such as air temperature, wind, and rainfall. For this purpose, the Schmidt index (Wang et al., 2019, Wang et al., 2012, Winder and Schindler, 2004), RWCS (Relative Water Column Stability) (Zhang et al., 2021b, Zhu et al., 2013), RTRM (Relative Thermal Resistance to Mixing) (Pu et al., 2020, Wetzel and Likens, 2000), and other global indexes have been widely adopted. The thermocline, which is observed as a region of sharp changes in temperature, separates the epilimnion and the hypolimnion, and significantly impacts water movement and vertical substance translocation (Hachaj and Szlapa, 2017). At present, the gradient criterion is mostly used to identify these thermal layers, as it distinguishes a region with a temperature gradient greater than a specific threshold as the thermocline, and the remaining two layers are then determined. This method is concise and practical, but the threshold is given empirically and varies from 0.2 $°C·m^{-1}$ (Yang et al., 2020a, Zhang et al., 2014a), 1.0 $°C·m^{-1}$ (Hadley et al., 2014, Wang et al., 2012), and 2.0 $°C·m^{-1}$ (Huang et al., 2016, Coloso et al., 2011) with the environment.

Section 2.3: It would improve the readability of the materials and methods section, if the different data analyses were more clearly linked with specific hypotheses which already stated in the results section.

Thank you for your instruction. Firstly, to clarify hypotheses, we have revised the aims of the study in the Introduction, as follow: "Therefore, the purpose of this study was to (i) diagnose the stratification degree and identify the stratification characteristic in the Nakdong River, (ii) examine the relationships between stratification degree with hydrometeorological variations and vertical nutrients patterns, and (iii) identify the phytoplankton changes in community composition and vertical cell distribution associated with stratification variability. We hypothesized that the river stratification would have different relationships with environmental variables and effects on phytoplankton between seasonal and diel scales, then we analyzed them separately. By including water quality and hydrometeorological parameters, mediation strategies for what? were discussed based on the results."

Secondly, we have extensively revised the Materials and Methods to improve its readability by highlighting the hypotheses for each analysis.
As for the first aim, three thermal stratification indices were selected and calculated. To underscore this, we have revised the sentence as "To identify the existence and intensity of thermal stratification in the Nakdong River, three indices… " (from line 91 in Section 2.2).
As for the second aim, we have revised the phrase, "To examine the correlation structure among environmental variables including the stratification indices," for the link with principal component analyses (PCA).
As for the third aim, the sentence for the canonical correspondence analysis (CCA; replaced into RDA in the revised MS) were changed to "To delineate the effect of

significant environmental drivers including the stratification indices on the vertical structure of phytoplankton community,".

Additionally, we have changed the subtitle for Section 3.1. in Results from "Stratification indices" to "Stratification patterns".

Section 3.1: temporal variations in the stratification indices are investigated, but why are the authors interested in the scales of variation? What do they expect? This is one of many examples, where the formulation of a hypothesis would improve the storyline. Are the authors expecting that short-term stratification will have a different ecological mechanism or consequence from lake stratification which persists longer?

Yes, we investigate how long did the river stratification persists and compare the duration with other stratification cases in the Discussion section as written below.

"An earlier onset of thermal stratification can lead to an increase in the spring peak biomass of phytoplankton which can lower summer biomass of zooplankton (George and Taylor, 1995). Consequently, phenological change in plankton seasonality can be influenced by the change in timing of stratification (Thackeray et al., 2008; Winder and Schindler, 2004)".

Fig 4: why are the authors presenting additional information on the thermoclines and their vertical variations? It would be easier to read if the authors formulated a hypothesis about how the diel variation of the thermoclines affect the vertical distribution of phytoplankton cell in Fig 8 and then investigate these.

Thermocline depth and its vertical migration are known to control cell density and vertical distribution of the phytoplankton (Santos et al., 2015). We revised the Introduction and Discussion.

Section 3.2: the first paragraph summarizes the changes over seasons and sites in the multiple parameters which were later analyzed against the stratification indices. From reading, it is not clear why all this information (and with the standard error of detail) is presented. Parts of the paragraph are trivial and the text could easily be reduced substantially (e.g. the two first sentences could be removed).

The research questions and corresponding results are like below.

H1. Is the river in eutrophic or physically calm status during the stratification period?: R1. Trophic status/ hydrometeorological condition

H2. Does the stratification affect the phytoplankton via vertical difference in the water quality within the water column?: R2. Kruskal Wallis test on the water quality at various water depths

The text had been shortened as written below.

"Table 2 summarizes the environmental conditions of the lower Nakdong River for the five months. Though several morphological (i.e., depth) and hydrological (i.e., flow rate and water level) parameters were site-dependent, all three sites were highly eutrophic based on nutrient concentrations and chlorophyll a concentration. All the water quality

variables were not significantly different among the three water depths of 0 m, 3 m, and bottom (Kruskal Wallis test, p > 0.05, n=15). However, the diel survey in August showed that water temperature and chlorophyll a varied significantly with depth (Kruskal Wallis test, p < 0.01, n=18). Dunn post hoc tests revealed that differences between WT0, WT3, and WTB were all significant, but ChlB was only significantly different from Chl0 and Chl3 (p < 0.05, n=18)."

Section 3.3: relationships between phytoplankton assemblage and multiple environmental factors including the stratification indices are investigated. It is described that the diel CCA showed a positive relationship between air temperature and cyanobacterial density. The authors must draw a conclusion by combining the PCA results, which showed a strong relationship between chlorophyll a and the stratification indices.

We recognized that both PCA and CCA depicted the relationship between the stratification indices and phytoplankton abundance differently at the diel scale. We solved this by replacing CCA into RDA assuming a linear response in phytoplankton community against an environmental gradient as we assumed linear among variable relationships in PCA. RDA has been widely used to describe changes in phytoplankton community in stratified freshwater ecosystems (Becker et al., 2009; Xiao et al. 2011; Zhou et al., 2016). RDA showed a high association of cyanobacteria to RWCS. RDA maintained high model significance (p<0.01) and explained more variation in phytoplankton data compared to CCA by 16.8%.

Fig 8. I suggest the authors to present cyanobacterial cell density, which was used in the CCA analyses in Fig 7. This may give a reason for the different stratification-phytoplankton relationships between the PCA and CCA.

Thanks for this comment. We replaced chlorophyll a concentration into cyanobacterial cell density in Fig. 8 and overlaid it with the depths of thermoclines.

Section 4.2: 'The PCA ordinations revealed that thermal stratification is one of the most important drivers of water environments in the Nakdong River, largely accounting for their seasonal and diel variations'. What do the authors mean by this?

Each ordination axis is a linear combination of all explanatory variables. The PCA returned the ordination axes corresponding to the directions of greatest variability within the dataset (meteo-hydrological variables, water quality, and stratification indices). As the sites and seasons (or times) were ordinated along these axes, and the stratification indices had higher loading to the axes. Therefore, we concluded that 'thermal stratification is one of the most important drivers of water environments in the Nakdong River, largely accounting for their seasonal and diel variations'

Reference

Kirillin, G., & Shatwell, T. (2016). Generalized scaling of seasonal thermal stratification in lakes. Earth-Science Reviews, 161, 179-190.

Thank you for the reference.

---

## Author Comment (AC2)

**Response to referee #2 (our answers in blue):**

I recommend the authors consider the following comments and develop the manuscript in a global context and perspective.

General comments

This manuscript focuses on river stratification, which was not common in the past but has become frequent due to human river modification and climate changes. The authors traced the causes of stratification and its impact on phytoplankton in the middle of the Nakdong River in South Korea after a river modification. They conducted vertical profiling seasonally and diurnally between 2017 and 2018, and diagnosed the stratification degree using three stratification indices. They also implemented various analyses to reveal the effect of stratification on the river microbial ecology, including associations between different water quality parameters and phytoplankton assemblages. The authors concluded that: 1) The summer stratification in 2018 exceeded the stratification thresholds generally accepted in many aquatic environments, and 2) Diel variations in the stratification intensity and thermocline depth were distinctive from the typical stratification of lakes or dams.

As a reviewer, I concur with the authors in their conclusions. I see several interesting points of the study, which are unfortunately not stressed at the current state of the manuscript. River utilization patterns become complex worldwide, and stratification often occurs in lotic systems. Most importantly, the structure and function of the aquatic ecosystem of the Nakdong River must be shifted to a new phase due to a large-scale river transformation project (as the authors illustrated). I ask the authors to relate the stratification occurrence and the river project; hence the readers may find helpful information comparable to their respective study sites. The authors provided a lot of analytical results, but they are not well related, nor successfully emphasize their findings. Consequently, the manuscript must be restructured largely. In my reading, I also found several unclear sentences. I have listed some of these under my minor comments. Please ask a native English speaker in the related scientific field to revise the English writing.

I hope that the authors' findings will be successfully conveyed to the readers of the journal Biogeosciences. I believe that an appropriate revision is necessary for this manuscript to meet the goals pursued by the Journal. Please revise the manuscript by carefully reflecting on this general and the following specific comments.

We would like to thank Reviewer#2 for the positive and constructive review. We have carefully responded to each comment/suggestion and did our best to improve the manuscript accordingly. We clarified and simplified the paper all along: We re-wrote the methods section to make it easier for the reader to follow. The diel stratification was discussed more thoroughly. All co-authors edited the letter and the manuscript, with additional English editing by a native speaker collaborator. We believe that the paper flows better now, making it easier for the reader.

Specific comments

- Title

Title of the manuscript may be reconsidered as the authors revise the manuscript.
Please contact your Editor of the journal the possibility.

Thank you. We will ask the journal about this.

- Abstract

Line 11. In the discussion part, the authors referenced the researches on diel river
stratification and response of phytoplankton. However, in this line they suggested the
studies are "lacking". Please revise it.

Thanks for this comment. We revised it.

Lines 23-25. If the authors accept the general comment, the abstract must be rewritten.
Especially, please stress your new findings or suggestion in this part.

We will revise the abstract by stressing three novel findings written below.

Firstly, we found that the seasonal shifts in phytoplankton community structure were
either insensitive and showed a limited response to the stratification indices. Summer
cyanobacterial bloom intensity, here cell abundance and accumulation into the surface
water, was positively affected by the diel variations in the stratification indices and
thermocline.

Secondly, *Microcystis* dominated during the stratification period and diel variation in their
cell densities at the surface were mostly affected by the stratification extent than other
environmental variables (nutrients, meteorological factors).

Thirdly, Phytoplankton is known to change its vertical position in stratified lake where the
waterbody is vertically segregated (Becker et al., 2009). Vertical position of
phytoplankton is known to be attributed to unequally distributed materials such as
nutrients (Mellard et al., 2011). However, in this study, cell densities of bacillariophyta,
chlorophyta, and cyanobacteria were vertically uneven, even though there was no
significant difference in water quality variables. This implicates weak stratification would
be enough to impact freshwater ecosystem and many rivers gone through artificial flow
regulation or impoundment could be affected.

To highlight these findings as novelty of our study, we have extensively revised the
Discussion and Conclusion part.

- Introduction

Lines 40-43. Differentiation of stratification characteristics in rivers from the typical lake stratification would be one of the main conclusions of this study. Moreover, I think these lines must be restructured, in relation to the river project the Nakdong River has experienced.

The first two paragraphs including lines 40-43 were merged and the importance of river stratification was highlighted. To related the river project in the Nakdong River, influences of river alteration on thermal structure were supplemented.

Lines 44-50. This is more appropriate for the study site description part so please move the sentences. Instead, explain why the Nakdong River stratification problem is import in global context.

Thanks for this comment. The paragraph was moved into the site description part.

- Methods

Line 60. It is strongly recommended the study site establishment strategy: i.e. I hope to understand why the authors set their sites there. In the results section, the different river morphologies or distances from the weirs brought different spatial variations in the stratification indices. This must be also related to the objectives of the study.

After the project, severe water quality deterioration and excessive cyanobacterial proliferation previously restricted to downstream areas of the river are now frequently reported in midstream areas (Park et al., 2021). These effects are often considered a consequence of the formation of stratification. Therefore, we focused into the mid-reach of the river expecting stratification. The basis for the site selection was added in the Methods section as written below.

"We selected three study sites in the mid-reach of the river based on the relative location to the weirs. St.1(upstream) located between two weirs and St.2(midstream) located just below the weir nearest the estuarine barrage. St. 3(downstream) located farthest from the weir."

Lines 113-139. It is strongly recommended to reformulate this part. I realized that they first examine descriptive characteristics of the study sites, then stratification events were investigated with the application of three indices. Then they applied several multi-variate statistical analyses such as PCA and CCA. Please detail the purpose and differences these analyses to facilitate the readers' understanding. I suggest the authors to reinspect the uses of similar analyses.

We recognized that both PCA and CCA depicted the relationship between the stratification indices and phytoplankton abundance differently at the diel scale. We solved this by replacing CCA into RDA assuming a linear response in phytoplankton community against an environmental gradient as we assumed linear among variable relationships in PCA. RDA has been widely used to describe changes in phytoplankton community in stratified freshwater ecosystems (Becker et al., 2009; Xiao et al. 2011; Zhou et al., 2016). RDA showed a high association of cyanobacteria to RWCS. RDA

maintained high model significance (p<0.01) and explained more variation in phytoplankton data compared to CCA by 16.8%.

We will keep using two multivariate analyses (PCA and RDA) based on the reasons below. To clarify the different purposes of each analysis, we have revised the phrase, "To examine the correlation structure among environmental variables including the stratification indices," for the link with principal component analyses (PCA).
The sentence for the redundancy analysis (RDA) were changed to "To delineate the effect of significant environmental drivers including the stratification indices on the vertical structure of phytoplankton community,".

We used RDA because it decomposes the variation in phytoplankton variables is into variation related to environmental variables represented by constrained axes. While in the case of unconstrained ordination (i.e., PCA) the information we are interested is mostly about the configuration of samples and species in the ordination diagram, the relative importance of individual ordination axes (measured by their eigenvalues) and ecological interpretation of ordination axes, in the case of constrained ordinations (i.e., RDA) we are more interested in the effect of environmental variables on species composition, namely in the amount of variation these variables explain and whether this variation is significant or not (see Explained variation and Monte Carlo permutation test), which of the available environmental variables are important to explain the variation of studies community (Forward selection) and how to partition the variation explained by different variables or different sets of variables (Variation partitioning).

Throughout the method part: Please keep methods the authors used in order of the analysis sequence, hence should be linked to the sequence of results exhibition.

Thank you. We will realign the M&M in accordance to the sequence in the Results section.

- Results

Please revise the Results part as the authors are asked to rewrite the methods part.

Thank you.

Lines 153-160. The authors diagnosed stratification in the river using three stratification indices. They presented much information of seasonal and spatial variation in the stratification for the respective stratification indices into the Figure 2. From reading, it is not clear which information they think the most important and relevant for phytoplankton.

In the Section 3.1 (Lines 153-160), we would like to identify 1) the seasonal and spatial variations and 2) the onset and end of the river stratification. However, we found little response in the seasonal shift in phytoplankton community composition as shown in Figure 7a. We had discussed the reasons for the limited response of phytoplankton to stratification in the Nakdong river as below.

"blooms of the cyanobacteria *Aphanizomenon* during fall-winter tolerant for water temperature in the range of 5–15 ℃ in the river (Ryu et al., 2016) hindered the detection of a significant relationship between stratification and cyanobacterial density in a seasonal scale."

Moreover, importance of stratification timing on plankton seasonality and peak abundance (George and Taylor, 1995; Thackeray et al., 2008; Winder and Schindler, 2004) was added in the Discussion.

Lines 166-174. Same comment. In the Discussion, it is explained that frequency of destratification and timing of the maximum stratification are important for cyanobacterial proliferation. Timing of the maximum stratification and depth of the thermocline were investigated. However how would they affect phytoplankton was not rigorously discussed in the Discussion.

We had discussed the inhibited cyanobacterial proliferation by destratification on diurnal basis. "Diel fluctuations in their biomass with high values of chlorophyll a in the surface water which did not persist as the thermocline weakened and deepened were primarily associated with stratification indices compared to other variables, such as water temperature and nutrients (Fig. 5b). According to Webster et al. (2000), whether or not the river water column is mixed at least once on a diurnal basis has great significance for the distribution of slowly floating Cyanobacterial genera, Anabaena. If the population was uniformly dispersed through the water column at sunrise, then during daylight hours, the population would not accumulate significantly into the near-surface euphotic zone where photosynthesis occurs". We will supplement more discussion for the effects of timing and thermocline depth.

Section 3.3. I concur with many comments already made by Referee #1. In addition to and supporting suggestions by Referee #1, I would recommend that the authors to identify the roles of diel variation in stratification and thermocline depth using the figure 7and 8 for the novelty of the study.

*Microcystis* dominated during the stratification period and diel variation in their cell densities at the surface were most closely related to the stratification indices. Similar result regarding the inhibiting effect of daily based destratification on Cyanobacteria was found in a temperate river (Webster et al, 2000) with our results would expand the Cyanobacterial genera affected by stratification from *Anabaena* to *Microcystis*, which occur in more freshwater ecosystems.

Lines 225-236. Phytoplankton distribution seems well responded to the stratification event (only by Figure 6); however, its corresponding text does not clearly explain. Please rewrite this part.

The research questions in Lines 225-263 are like below.

H1. Seasonal change in the community composition?: R1. Domination by cyanobacteria in stratification period (summer)

H2. Which phylum exhibits vertical density variation?: R2. All phyla were abundant at the specific water depths at diel scale.

Lines 225-263 were followed by constrained ordination analyses which aimed to test whether these features of phytoplankton were affected by river stratification.

The revised paragraph (Lines 225-263) is like below.

"The phytoplankton community showed a seasonal shift at the phylum level (Fig. 6). Massive cyanobacterial blooms were observed in November, August, and September, with high cell densities concentrated in the surface waters. The blooming cyanobacterial genera varied by season, with *Microcystis* for August, the strongest stratification period, and *Aphanizomenon* for November and September, when the waterbodies mixed well and their proliferation was independent of stratification (Table S2). In May, Bacillariophytes dominated, and in March-April and September, co-domination of Bacillariophytes and Chlorophytes was detected. Unlike cyanobacteria, the cell densities of bacillariophytes and chlorophytes showed less vertical variation. For all the phytoplankton phyla, cell densities were not significantly different among the three water depths of 0 m, 3 m, and bottom (Kruskal Wallis test, $p > 0.05$, n=15). However, the diel survey in August showed that all phyla showed significant differences in cell densities among water depths (Kruskal Wallis test, $p < 0.05$, n=18) (Table S3).   Dunn post hoc tests revealed the water depths with significantly higher cell densities for each phylum: Cya0 and Cya3, BacB, and ChlB ($p < 0.05$, n=18)"

Lines 242-249. Relationships between phytoplankton assemblage and environmental variables including several stratification indices were analyzed. In interpreting the results, do they assume the response of phytoplankton in an "assemblage-scale" or a linear relationship in an individual phytoplankton phylum? Also, please highlight and distinguish these finding from the similar researches in the Discussion section.

We would like to describe the response of phytoplankton in a "community-scale" rather than individual phylum. We assumed that three phyla (bacillariophyta, cyanobacteria, chlorophyta) at three different depths constitute a community at a sampling time. Based on the assumption, we performed CCA (replaced into RDA in the revised MS) to identify changes in the phytoplankton community composition and impact of environmental drivers on vertical cell distribution. To clarify this assumption, we have added the sentence, "We assumed that three phyla (bacillariophyta, cyanobacteria, chlorophyta) at three different depths constitute a community at a sampling time" in the Methods and Material section.

- Discussion

Subsection 4.1. They entitled this part as "the characteristics of the Nakdong River", however the Nakdong River circumstance was not well discussed. The spatial difference of the stratification indices was found between the study sites. They should explain how the different river morphologies or distances from the weirs brought these consequences. For example, the authors may relate future mitigation strategy for cyanobacterial proliferation to stratification patterns.

Overestimation of the stratification in the midstream by S compared to the other indices indicates that the waterbody of midstream required more turbulent energy to mix than expected for its entire stability and thermocline strength, especially in August. Moreover,

the hydrological factors and wind velocity exhibited greater spatial variability, indicating spatial heterogeneity in susceptibility to thermal stratification formation. This makes the upstream site more favorable for stratification in terms of its lower flow rate, lower wind velocity, and higher water level.

Moreover, they only stressed that the river had different stratification characteristics than the typical lake system by comparing the stratification indices and behaviors of stratification to referenced lakes. Finally, they concluded that these characteristics are important in shaping the aquatic ecosystem. Please relate the results in the manuscript to the storyline. Please suggest how unique or general the stratification in the Nakdong river is by comparing to other river stratifications.

One comparable river stratification is from a regulated river Saar (Engel and Fischer et al., 2016). Stratification increased the phytoplankton abundance and vertical differences only when the abundance is low in river Saar. However, in this study, summer stratification had positive association to the phytoplankton abundance and vertical differences when there was abundant cyanobacterial cell density.

Section 4.2. They expect that summer stratification would become more intensive in the Nakdong river based on the dependence of the stratification indices on the meteorological variables. It would be more persuasive if evidences were provided properly. For example, they can show the highest record of air temperature of the study site in 2018 or extended years then compare it with the air temperature at the date of surveys. This will tell how much the stratification would further intensify as it reaches the middle of summer and how the stratification would be recurrent in every summer.

We added air temperature data that stratification observed in this study seems to recurrent in the river basin. We have extensively revised the Discussion and Conclusion part.

Section 4.3. The authors are asked to discuss their results with the references shown in the texts. If the authors accept my suggestion shown above, the texts in this section must be largely restructured.

Section 4.3 includes highlighting results of the study as written below. We will restructure this section to better separate our results from discussion.

1) A phylum-level shift between seasons, the major variation in the phytoplankton community, was not associated with the stratification indices.
2) Two harmful cyanobacterial species, *Microcystis wesenbergii* and *M. aeruginosa*, were enhanced by summer stratification in terms of their abundance and cell accumulation in the surface water.
3) Diel fluctuations in their biomass with high values of chlorophyll a in the surface water which did not persist as the thermocline weakened and deepened were primarily associated with stratification indices compared to other variables, such as water temperature and nutrients.

- Conclusion

Is there really a need for a concluding section? This section is basically a summary and not a terse concluding paragraph. It needs to be shorter and highlight the novelties of the study. If this cannot be done, then it is not needed.

We improved the conclusion section by starting with a short introductory and overview of the results to better separate sections and to guide the reader.

Technical comments

Lines 57-62. As aforementioned, the sentences of Nakdong River explanation may be merged to this part. Please do not forget to clarify the Nakdong River characteristics, comparable to other rivers in the world.

Sentences describing the morphological changes in the Nakdong river were merged into the Study area section.

Line 70. Please explain your diel monitoring strategy in detail.

Thanks for this comment. The sampling strategy had already been explained in L67-70 and it was identical for the diel monitoring as stated in L70-71. The whole paragraph will be revised for clarity.

Line 76. The authors examined phytoplankton species data from their diel monitoring, but here they did not explain how they monitored, identified, and enumerated.

Thanks for this comment. It was answered together at the previous comment on Line 70.

Figure 1 (the map). Please add a small map showing the East Asian region to understand where the river is located.

Thanks for this comment. It was corrected in the revised MS.

Lines 97, 102. I don't think the in-line equations are necessary, because the authors already showed them in Table 1. Please consult the journal style.

Thanks for this comment. All the in-line equations were removed.

Line 103. Acceleration due to gravity ïƒ gravitational acceleration

Thanks for this comment. "Acceleration due to gravity" was changed into "gravitational acceleration".

Table 1. Please check whether the authors used those threshold values as the meaning of a magnitude or intensity that must be exceeded for stratification. Why they provided two values for the respective indices? In my reading, the terms "low- and high-threshold" could be misunderstood to mean the lowest and the highest values of each of the indices reported previously.

We use the term "threshold" as the general meaning.

Also, the authors did not provide equation of "Maximum temperature gradient" in the table, and what is "many" right after the name of this index?

Thanks for this comment. A reference "Lampert & Sommer, 1997" and an equation "Max = $(\Delta T/\Delta d)_{max}$" was added for the Maximum temperature gradient in the Table1.

Line 148. Please show the version numbers for respective R packages the authors used.

Thanks for this comment. Version numbers for respective R packages were added.

Lines 155-157. Different results between the stratification indices were explained. Please move it to the Discussion section.

Thanks for this comment. The sentence was moved to the discussion.

Lines 181-190. I think this descriptive part may appear in the first part of the results.

We would like to keep the position. Lines 181-190 describe the overall environmental conditions of the river concluding that the river was in a highly eutrophic and hydro-meteorologically calm status. To guide readers' intuitions about ecological consequences of stratification occurrence in a calm and eutrophic waterbody, we arranged the results section in the order of stratification pattern (section 3.1), environmental conditions (3.2), and phytoplankton community (3.3).

Table 3. This would be appropriate for supplementary materials.

Thanks for this comment. The Table 3 was moved into the supplementary materials.

Lines 259-267. Please move this paragraph to the previous diel pattern explanation part, and concentrate on Microcystis distribution. Also, please explain what data were used to calculate each average value.

The mentioned paragraph is irrelevant with the diel variation in phytoplankton community, but describes the spatial difference in *Microcystis* bloom intensity attributed to the spatial differences in stratification extent. However, we recognized that there is little need for section 3.4 dedicated only for *Microcystis* then we merged Lines 259-274 into section 3.3 (phytoplankton community) after shortening the text.

Figure 8. Please check the parentheses.

Thanks for this comment. The parentheses were removed from the figure 8.